# Does Stocking Density Affect Growth Performance and Hematological Parameters of Juvenile Olive Flounder *Paralichthys olivaceus* in a Recirculating Aquaculture System?

**DOI:** 10.3390/ani13010044

**Published:** 2022-12-22

**Authors:** Junhyuk Seo, Jeonghwan Park

**Affiliations:** 1Advanced Aquaculture Research Center, National Institute of Fisheries Science, Changwon 51688, Republic of Korea; 2Division of Fisheries Life Sciences, College of fisheries science, Pukyong National University, Busan 48513, Republic of Korea

**Keywords:** recirculating aquaculture system, olive flounder, stocking density, growth performance, animal welfare

## Abstract

**Simple Summary:**

Stocking density is a key factor for animal welfare in aquaculture, as it can cause stress in fish. However, a low stocking density could reduce feed competition among cultured fish, resulting in a decrease in feed intake and, consequently, productivity. To assess both animal welfare and productivity in aquaculture, the appropriate stocking density for the target species must be investigated. Based on the growth performance, hematological response, and whole-body composition results in the present study, the maximum stocking density should not exceed 23 kg/m^2^ for juvenile olive flounder (60–180 g in individual weight).

**Abstract:**

Olive flounder *Paralichthys olivaceus* is a representative culture species in South Korea. Recirculating aquaculture systems (RASs) have received increased attention because they can provide sustainable and environmentally friendly productivity. However, to maintain economic sustainability, the system generally requires high productivity, achieved through a high stocking density, which compromises animal welfare. The reduction in growth based on the stocking density may be due to the social hierarchy resulting from the growth suppression of subordinate individuals. Species, size, culture systems, and other management regimes can affect the social hierarchy. Therefore, a more practical approach must be taken to adjust the stocking density for a particular fish species and fish size in a specific culture system. This study investigated the effect of stocking density on juvenile olive flounder in an RAS. Juvenile olive flounder (61.0 ± 0.3 g) were initially stocked at 3.29, 4.84, 7.14, and 8.56 kg/m^2^ (T1, T2, T3, and T4, respectively). After 8 weeks, growth performance, in terms of feed conversion, specific growth rate, and daily feed intake rate, was measured. In addition, the blood levels of insulin-like growth factor (IGF-1), growth hormone, glucose, glutamic oxaloacetic transaminase, and glutamic pyruvic transaminase and the whole-body composition were evaluated as stress indicators. Growth performance increased as stocking density increased, but fish at the highest stocking density showed signs of growth reduction toward the end of the experiment. In addition, as stocking density increased, IGF-1 decreased, and cortisol increased. The whole-body protein level was significantly lower in T4 compared with the other treatments. Olive flounder seemed to tolerate a stocking density up to 20.16 kg/m^2^. Based on the growth performance, hematological response, and whole-body composition results in the present study, a final stocking density of 20 kg/m^2^ (from the initial stocking density of 4.84–7.14 kg/m^2^) may be desirable for juvenile olive flounder in an RAS for fish weighing ~60–180 g.

## 1. Introduction

Olive flounder *Paralichthys olivaceus* is one of the most popular aquaculture species in South Korea; it is mostly cultured in flow-through systems. However, in recent years, its productivity has gradually declined due to increasing disease outbreaks associated with the pollution of the coastal environment. Recirculating aquaculture systems (RASs) have received increased attention because they are highly independent of the external environment and can cope with this problem [1,2,3]. RASs recycle used water through various water treatment devices and can provide many benefits to farmers by consuming relatively little water. They can allow a higher degree of control of the internal climate and water environment to promote year-round growth and the intensification of culture animals and drain pollutants to increase environmental sustainability [4,5,6,7,8]. More importantly, such systems can enhance biosecurity by reducing the risk of infection from external sources of disease and pollution. However, these systems also require sufficient knowledge of management and feeding regimes. Failure to maintain an adequate internal environment may result in mass mortality and delayed growth due to biological stress and diseases associated with intensified culture conditions [9]. Therefore, healthy and successful production requires the adequate management of animal welfare by reducing stressors to the culture animals [10,11,12,13]. Animal welfare refers to an individual’s condition in relation to adaptation to the biological, behavioral, and psychological aspects of the environment. When animals cannot cope with a given environment, their welfare is compromised, and their health conditions are degraded [14,15]. The concern for animal welfare has also increased in relation to aquatic animals and has recently become one of the priorities of the aquaculture industry.

Various factors affect animal welfare, including handling, transportation, diseases, feed, and slaughter methods. Stocking density is one of the major factors affecting animal welfare, as it influences stress, aggression, and feed intake [10,16,17]. Consequently, it determines the growth rate, survival rate, feed conversion ratio, and productivity [18,19,20]. It has been reported that an excessive stocking density can act as a stressor for ayu *Plecoglossus altivelis*, rainbow trout *Oncorhynchus mykiss*, gilthead seabream *Sparus aurata*, and halibut *Hippoglossus hippoglossus* L. and inhibit growth [10,21,22,23,24]. However, less feed competition among cultured fish has been reported at low stocking densities, resulting in a decrease in feed intake [23,25]. To assess animal welfare and productivity in aquaculture, the appropriate stocking density for the target species must be investigated. Studies of animal welfare in relation to stocking density have been well-documented for some species in RASs, but not for olive flounder [26,27,28]. Therefore, this study investigated the effect of stocking density on productivity and stress when rearing olive flounder in an RAS.

## 2. Materials and Methods

### 2.1. Experimental System Design and Culture Environment

The experiment was conducted in a seawater RAS comprising twelve 700 L square fish tanks; a drum filter (80 µm, NT-PM-20, Netech, Guangzhou, China); a biological filter tank; an ultraviolet (UV) sterilizer (80W, HS-50, Hansung-uv, Seongnam, South Korea); a pure oxygen generator (O2MOS-7LC, OXUS, Gapyeong, South Korea); an oxygen dissolving device; a cooler; and a heater (Figure 1). The biological filtration tank comprised two connected 1 m³ tanks with a water level at 700 L and a 70% filtration material filling rate; the system was sufficiently aged before the experiment. The total volume of the culture systems excluding water in the pipes was about 10,000 L; the recirculation rate of the system was 24 cycles per day; and the daily make-up water was maintained at 10%, including drum filter backwashing water. The recirculation rate was adjusted using a by-pass connected to a pump (290 LPM, PU-S1700M, Wilo, Dortmund, Germany), and one tank identical to the biological filtration tank was configured as a reservoir tank (or make-up tank).

During the experiment, the dissolved oxygen in all culture tanks amounted to 11.1 ± 0.8 mg/L, the pH was 7.77 ± 0.10, the total ammonia nitrogen amounted to 0.78 ± 0.43 mg/L, the nitrite nitrogen amounted to 0.51 ± 0.21 mg/L, and the nitrate nitrogen was maintained at 26.1 ± 3.6 mg/L. The dissolved oxygen and pH were maintained at >8 mg/L and pH 7 using pure oxygen and sodium hydrogen carbonate, respectively. Measurements were made using a water quality multimeter (HQ40D, Hach, Loveland, CO, USA) with a dissolved oxygen sensor (LDO101, Hach) and a pH sensor (PHC101, Hach). Total ammonia nitrogen and nitrite nitrogen were determined separately using the Indo-phenol method and the Diazotation method, respectively, and measured with a spectrophotometer (UV-3300, Humas, Daejeon, South Korea). Nitrate nitrogen was determined using the cadmium reduction method (Hach Method 8039) and measured with a spectrophotometer (DR900, Hach, Loveland, CO, USA).

### 2.2. Culture Management and Hematological Analyses

Olive flounder (initial weight of 61.0 ± 0.3 g per individual) was cultured in the seawater RAS at four stocking densities, namely 50 (Treatment 1, T1); 75 (Treatment 2, T2); 100 (Treatment3, T3); or 125 (Treatment 4, T4) juvenile olive flounders (Table 1).

Fish were acclimatized for 1 week before the start of the experiment and fasted for 3 days, including before and after accommodation, during transportation to the experimental system. We applied a 12 h photoperiod (light from 09:00 to 21:00 and dark from 21:00 to 09:00) throughout the entire experiment. The water temperature was maintained at 25 ± 0.5 °C with a cooler and heater, and a certain amount of fresh water was added to maintain salinity at 33–34‰ and prevent it from increasing due to evaporation. The experimental fish were fed with a commercial extruded pellet diet (Daebong LF, Jeju, South Korea, 55% crude protein, 8% crude lipid) three times a day for 4 weeks after the start of the experiment, and two times a day for the last 4 weeks of the experiment. The leftover feed was collected and excluded from the amount of feed supplied.

At weeks 4 and 8, the total number of fish and growth performance were evaluated for each treatment. Fish fasted for 3 days, including before and after the measurements. Ten fish from each tank (a total of 30 fish for each treatment) were randomly selected, and individual weight and length were measured. Using the measured data, the feed conversion, specific growth, daily feed intake, survival rates, and condition factor were calculated using the following formulae.

Feed conversion = feed consumption/weight gain;Specific growth rate (%/day) = ((ln final fish weight–ln initial fish weight)/number of culture days) × 100;Daily feed intake rate (%/day) = specific growth rate × feed conversion;Survival rate (%) = (number of surviving fish/total number of initially stocked fish) × 100;Condition factor = (fish weight/length^3^) × 100.

To investigate how stocking density affected hematological measures, 15 fish were randomly selected from each culture tank, and blood was collected. During blood collection, fish were anesthetized using ethyl 3-aminobenzoate methanesulfonate (MS-222, Sigma-Aldrich, Saint Louis, MO, USA) to minimize handling stress, and blood was collected from the tail blood vessel using a syringe (1 mL) treated with heparin (Sigma-Aldrich). Blood components were separated by centrifugation (4 °C, 1500× *g*, 12 min), and plasma was removed and frozen at −70 °C until analysis (to prevent degradation). 

The separated plasma was used to examine the effects of stocking density on growth and stress factors in fish, namely growth hormone (GH), insulin-like growth factor-1 (IGF-1), cortisol, glucose, glutamic oxaloacetic transaminase (GOT), glutamic pyruvic transaminase (GPT), and superoxide dismutase (SOD). Glucose, GOT, and GPT were analyzed using an automatic clinical chemical analyzer (FUJI DRI-CHEM 4000i, Fujifilm, Tokyo, Japan) and FUJI DRI-CHEM slides (FUJI DRI-CHEM, Fujifilm). Enzyme-linked immunosorbent assay (ELISA) kits were used to measure GH (Cusabio, Wuhan, China); IGF-1 (Cusabio); and cortisol (Enzo, New York, NY, USA), and SOD activity was measured using a colorimetric assay kit (Dojindo, Washington, DC, USA).

Finally, 10 fish were randomly selected from each tank, and whole-body composition was determined. The Feed and Nutrition Research Center of Pukyong National University reported the moisture, raw meal, crude fat, and crude protein content of the fish.

### 2.3. Statistical Analysis

Statistical analysis was performed using SPSS Statistics Version 25.0 (Armonk, NY, USA). First, one-way analysis of variance (ANOVA) was conducted for each variable. Subsequently, Levene’s test was performed to test for homoscedasticity. If there was homoscedasticity, Duncan’s multiple range test was used for multiple comparisons. If there was no homoscedasticity, the Games–Howell multiple range test was used for multiple comparisons. For all analyses, *p* < 0.05 was considered significant.

## 3. Results

Table 2 shows the survival rates and growth results for each stocking density measured at 4 weeks (out of the total 8-week culture period). There were significant differences between the experimental groups in the specific growh rate and the condition factor, with relatively low growth rates in the group with the lowest stocking density (*p* < 0.05). The daily feed intake rate and feed efficiency were significantly higher at the highest stocking density, and the corresponding feed coefficient was lower at the higher stocking density (*p* < 0.05). There was no significant difference in the survival rate (*p* > 0.05). 

The final survival rates and growth parameters were measured at 8 weeks (Table 3). There were no significant differences in the condition factor among the treatments, but there was a significant difference in the specific growth rate between T4 and the other three treatments (*p* < 0.05). There were no significant differences between the experimental groups in the daily feed intake rate, feed coefficient, and feed efficiency (*p* > 0.05). Similarly to the 4-week measurements, the daily feed intake rate was higher for T3 and T4. In contrast to the 4-week measurements, we found a relatively high feed efficiency and a low feed coefficient for T1 and T2.

IGF-1 was not significantly different between T2 and T3, but the IGF-1 concentration in the blood decreased significantly as the culture density increased (*p* < 0.05, Figure 2). There was no significant difference in GH between the groups (*p* > 0.05, Figure 2). Cortisol was significantly higher at the highest stocking density (T4, *p* < 0.05, Figure 3). There was no significant difference between the experimental groups for glucose, GOT, and GPT (*p* > 0.05, Figure 4). 

Table 4 shows the results of the fish whole-body composition analysis. Moisture, crude fat, and crude ash were not significantly different among the groups (*p* > 0.05). On the other hand, crude protein was significantly lower in T4 (*p* < 0.05).

## 4. Discussion

During the 8-week experimental period, nitrogen levels in the form of total ammonia and nitrite were maintained below 1 mg/L in all culture tanks, with little effect on the growth and survival rate of olive flounder [29,30]. Water temperature, pH, dissolved oxygen, and nitrate nitrogen were also within the range suitable for olive flounder in all experimental groups [29,31,32]. Therefore, the results related to fish welfare indicators, such as growth, survival, and stress, were independent of the variation in water quality parameters between the fish tanks.

In this study, daily feed intake and growth performance indicators, such as the feed conversion ratio and specific growth rate, appeared to improve with increasing stocking density at week 4. Similar results have been reported in other studies with different fish species [33,34,35]. Some pond studies with juvenile largemouth bass *Micropterus salmoides* and hybrid bluegill *Lepomis macrochirus* × *Lepomis microlophus* also reported improved feed conversion at higher stocking densities [36,37,38]. European sea bass *Dicentrarchus labrax* (individual weight of 6.6 ± 1.1 g) reared for 168 days at stocking densities of 80, 165, 325, and 650 individuals/m^3^ exhibited a significantly higher growth rate and greater appetite in the highest stocking density group [33]. Senegalese sole *Solea senegalensis* also showed better growth performance with increasing stocking density and significantly higher feed intake at higher stocking densities [39]. The studies reported that the fish were subjected to less feed competition at lower stocking densities. This was consistent with the results for Arctic charr *Salvelinus alpinus*, which exhibited significantly lower foraging activity and, consequently, a lower daily feed intake rate at stocking densities of ≤15 kg/m^3^ [40]. In a study with continuous feeding at a fixed feeding rate for juvenile largemouth bass (individual weight of 9.04 ± 0.64 g) in a semi-closed recirculating system, fish showed no differences in feed conversion and specific growth rate for stocking densities of 15–40 kg/m^3^ over 60 days [41]. However, the feeding activity of the fish in this experiment was higher at higher stocking densities. At the two lowest stocking densities (15 and 20 kg/m^3^), juvenile largemouth bass often took more time to consume the feed offered at the same quantity because they foraged less. Although feeding regimes, culture systems, and other management factors could affect the growth performance of fish at different stocking densities, the results indicated that a higher stocking density could stimulate growth performance to some degree.

Other studies have indicated a negative relationship between stocking density and feed intake rate due to size differences and increased stress [19,42,43,44]. Reduced feed intake and increased stress could suppress feed conversion and the specific growth rate over time. Stocking density is a key factor for production efficiency and animal welfare in aquaculture and therefore needs to be adjusted appropriately throughout the production process. If stocking density is not properly adjusted in a timely manner, profitability will be compromised by either chronic growth reduction or disease outbreaks. In contrast to the results for the first 4 weeks of the present study, the daily feed intake rate and feed conversion appeared to be compromised as the stocking density increased during the final 4 weeks. In addition, the specific growth rate at the highest stocking density decreased when the final stocking density reached 23.14 kg/m^3^. Over time, the change in stocking density affected the feed intake rates and growth performance differently for each group. As fish grew, the groups with a lower stocking density were expected to experience greater feed competition, which was reflected in higher feed intake rates. In contrast, the groups with a higher stocking density were associated with more stress due to overcrowding toward the end of the experiment. The growth performance of Nile tilapia *Oreochromis niloticus* (individual weight of 141–152 g) also improved up to a stocking density of 50 fish/m^3^ (30, 40, 50, 60, and 70 fish/m^3^) but was impaired at the highest stocking densities in net cages for 90 days [35]. Stocker-size largemouth bass (individual weight of 36.7 ± 2.9 g) exhibited similar feed conversion and specific growth rates at stocking densities of 4.5–36 kg/m^3^ but impaired growth performance at stocking densities of 54–72 kg/m^3^ over 60 days [44]. Olive flounder is a benthic flatfish that exhibits somewhat similar characteristics to turbot. Irwin et al. [43] reported the reverse effect of stocking density (0.7, 1.1, 1.5, and 1.8 kg/m^2^) on the average weight of juvenile turbot *Scophthalmus maximus* (individual weight of 8.62 ± 0.06 g) reared in a flow-through system for 45 days. Another study found significantly slower growth and a lower live weight in turbot (individual weight of 41 g) reared in sea cages at different stocking densities (1.26, 2.44, 3.69, and 4.87 kg/m^2^) when the fish reached the stocking density of 15.4 kg/m^2^ [35]. In the present study, the specific growth rate of olive flounder also decreased when the final stocking density reached 23.14 kg/m^2^, which was consistent with the results for turbot, showing reduced growth performance at a stocking density of 23.3 kg/m^2^ [45]. However, another study reported that the growth performance of turbot was independent up to a stocking density of 68 kg/m^2^ [46], which was significantly higher than the values reported in this and the previously mentioned studies. Researchers often report that stocking density and growth rates are related, but the relationships between the two is not always uniformly positive or negative for a given species. Growth reduction as stocking density increases may be due to social hierarchy leading to intraspecific size differences in groups resulting from growth suppression of subordinate individuals by larger conspecifics. Species, size, culture systems, and other management regimes may also influence social hierarchy and determine the threshold for maximum stocking density in different ways. Therefore, adjusting stocking density for a particular fish species and size in a particular culture system represents a more practical approach.

Analyses of blood chemistry and whole-body composition can be used to determine the effects of stress on fish and their growth under various conditions. The results of these analyses in the present study appeared to confirm the results of growth performance. Although GH did not differ among the groups, IGF-1 levels showed an inverse relationship with stocking density. The group with the highest stocking density (final stocking density of 20.88 kg/m^3^) had significantly lower IGF-1 than the other groups. IGF-1 is a good candidate for measuring instantaneous fish growth, because it directly contributes to mediating cell proliferation and somatic growth, which is correlated with the specific growth rate in fish under different culture conditions that alter growth [47]. In the present study, the decrease in IGF-1 as stocking density increased could also imply increased stress at higher stocking densities. Studies in silver perch *Bidyanus bidyanus* and black bream *Acanthopagrus butcheri* have shown that the cortisol level is inversely proportional to the IGF-1 level [48]. This phenomenon seemed to be well reflected in the changes in cortisol with IGF-1 in the present study, because the cortisol level at the highest stocking density was significantly higher compared with the other groups. Previous studies have also reported that cortisol increases as stocking density increases, indicating a reduced growth rate for turbot in RASs, rohu *Labeo rohita* in cages, olive barb *Puntius sarana* in cages, common carp *Cyprinus carpio* in biofloc tanks, and Nile tilapia *O. niloticus* in biofloc tanks [49,50,51,52,53,54]. However, in other studies, the cortisol levels were not affected in African catfish *Clarias gariepinus* in tanks at a higher stocking density but decreased in Nile tilapia *O. niloticus* in biofloc tanks [55,56,57]. Glucose is a secondary stress indicator in fish and tends to be increased by stressors such as stocking density, transportation, and handling. Increased glucose levels were observed in olive barb in cages, gilthead seabream *Sparus auratus* in tanks, Senegalese sole and Chinese sturgeon *Acipenser sinensis* in RASs, common carp in biofloc tanks, and Asian seabass *Lates calcarifer* in cages [51,53,58,59,60,61]. In the present study, glucose, GOT, and GPT did not differ between the groups. 

Based on the whole-body composition assessment in the present study, the crude protein content in the group with the highest stocking density was significantly lower than that of the other experimental groups. In a study that evaluated the effect of stocking density on body composition in rainbow trout *O. mykiss*, the authors found a significant decrease in crude protein when the density reached 36 kg/m^3^ [61]. Meagre *Argyrosomus regius* with individuals weighing 9.15 ± 0.2 g reared in cages with different stocking densities (50, 150, and 250 fish/m^3^) showed significantly higher protein levels in groups with lower stocking densities [62]. The present study indicated that the intensive stocking density caused stress, resulting in the poor assimilation of feed nutrients for the groups with a high stocking density. The effects of stocking density on the levels of hematological factors and whole-body composition are species-, size-, and system-specific, and the optimal stocking density in relation to fish welfare should be determined by considering these factors. 

Although IGF-1 levels in T2 and T3 were significantly lower than in T1, the cortisol levels remained similar to T1 (the group with the lowest stocking density). While the levels of the other hematological factors did not differ among the groups, the whole-body protein level suggested the onset of a possible inverse effect on fish growth at the highest stocking density (T4). The specific growth and daily feed intake rates were impaired when the stocking density approached 23.1 kg/m^3^. Thus, olive flounder seemed to tolerate a stocking density up to 20.2 kg/m^2^, and this density should not be exceeded for 60–180 g individuals in an RAS. 

## 5. Conclusions

This study indicated that increasing the stocking density positively affected olive flounder growth. However, the hematological indices, such as the cortisol level, and the body protein composition suggested signs of stress, which compromises animal welfare. Collectively, based on the results for growth performance, hematological response, and whole-body composition, the desirable initial stocking density appeared to be in the range of 4.84–7.14 kg/m^2^, and the maximum stocking density should not exceed 20.2 kg/m^2^ for juvenile olive flounder weighing 60–180 g in an RAS.

## Figures and Tables

**Figure 1 animals-13-00044-f001:**
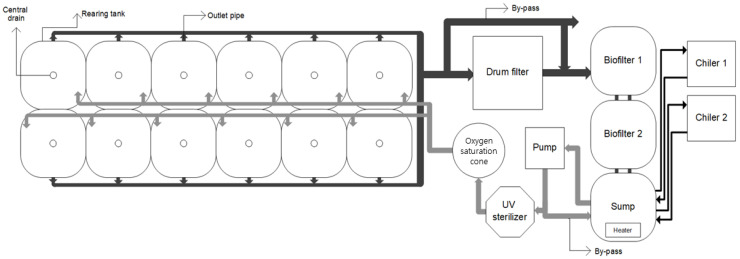
Schematic drawing of the experimental recirculating aquaculture system (dark arrows represent effluent, and gray arrows represent influent).

**Figure 2 animals-13-00044-f002:**
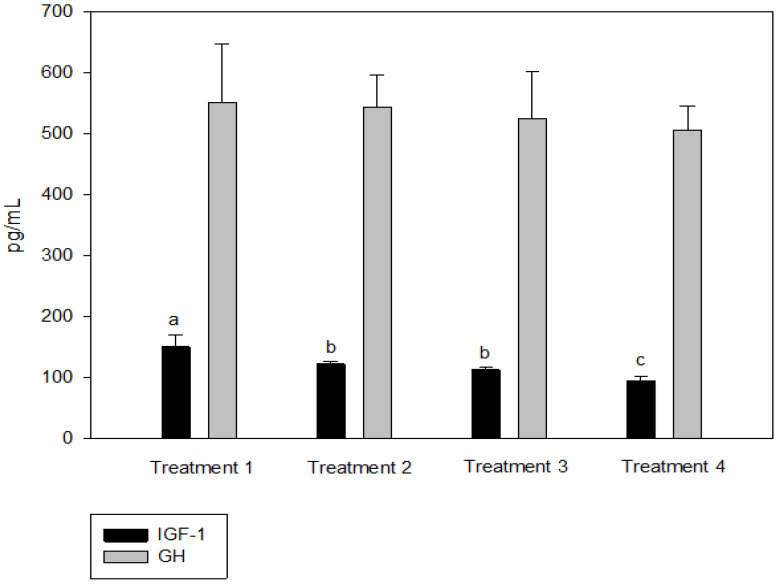
Insulin-like growth factor (IGF-1) and growth hormone (GH) levels of olive flounder *Paralichthys olivaceus* cultured at four different initial stocking densities. The data are presented as mean ± standard deviation. (Small letters “a, b, c” indicate a significant statistical difference between the treatments.)

**Figure 3 animals-13-00044-f003:**
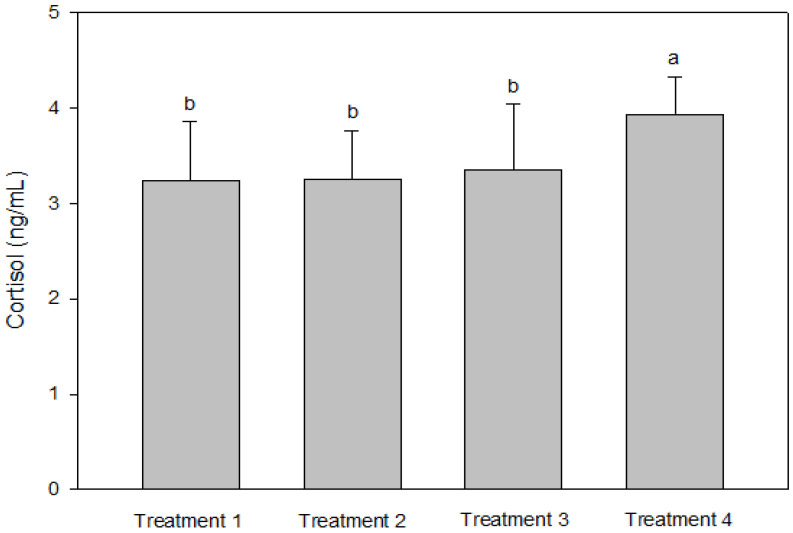
Cortisol levels of olive flounder *Paralichthys olivaceus* cultured at four different initial stocking densities. The data are presented as mean ± standard deviation. (Small letters “a, b” indicate a significant statistical difference between the treatments.)

**Figure 4 animals-13-00044-f004:**
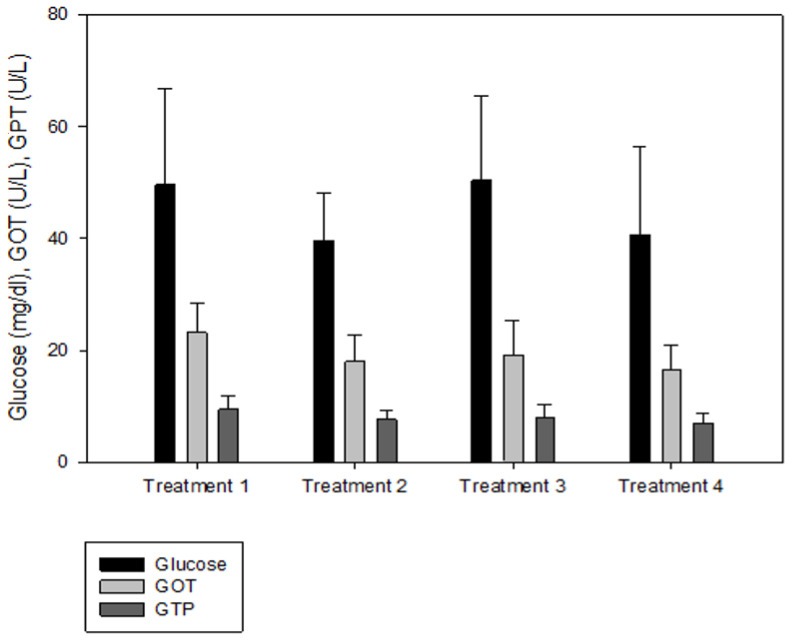
Glucose, glutamic oxaloacetic transaminase (GOT), and glutamic pyruvic transaminase (GPT) levels of olive flounder *Paralichthys olivaceus* cultured at four different initial stocking densities. The data are presented as mean ± standard deviation.

**Table 1 animals-13-00044-t001:** Initial stocking information for each treatment in the experimental recirculating aquaculture system to evaluate the effect of stocking density on the productivity and stress of olive flounder *Paralichthys olivaceus*.

Treatment	Treatment 1	Treatment 2	Treatment 3	Treatment 4
Number of fish	50	75	100	125
Total weight (kg)	2.97 ± 0.11	4.37 ± 0.11	6.44 ± 0.06	7.73 ± 0.17
Stocking density (kg/m^2^)	3.29 ± 0.13	4.84 ± 0.12	7.14 ± 0.06	8.56 ± 0.19

**Table 2 animals-13-00044-t002:** Growth performance of olive flounder *Paralichthys olivaceus* cultured at four different initial stocking densities for 4 weeks.

Parameter	Treatment 1	Treatment 2	Treatment 3	Treatment 4	*p*
Initial biomass (kg)	2.97 ± 0.11 ^d^	4.37 ± 0.11 ^c^	6.44 ± 0.06 ^b^	7.73 ± 0.17 ^a^	0.000
Initial stocking density (kg/m^2^)	3.29 ± 0.13 ^d^	4.84 ± 0.12 ^c^	7.14 ± 0.06 ^b^	8.56 ± 0.19 ^a^	0.000
Midterm biomass (kg)	4.25 ± 0.01 ^d^	7.97 ± 0.57 ^c^	12.79 ± 0.40 ^b^	14.88 ± 1.07 ^a^	0.000
Midterm stocking density (kg/m^2^)	4.70 ± 0.51 ^d^	8.83 ± 0.63 ^c^	14.17 ± 0.44 ^b^	16.48 ± 1.19 ^a^	0.000
Feed conversion	1.29 ± 0.27 ^a^	0.93 ± 0.04 ^b^	0.93 ± 0.03 ^b^	0.92 ± 0.06 ^b^	0.002
Specific growth rate (%/day)	1.07 ± 0.09 ^c^	1.78 ± 0.16 ^b^	2.00 ± 0.01 ^a^	1.93 ± 0.09 ^a^	0.000
Daily feed intake rate (%/day)	1.36 ± 0.19 ^c^	1.66 ± 0.21 ^b^	1.86 ± 0.05 ^a^	1.77 ± 0.04 ^a^	0.015
Survival rate (%)	98.7 ± 1.2	97.8 ± 3.8	98.7 ± 2.3	97.9 ± 2.0	0.950
Condition factor	0.84 ± 0.04 ^b^	0.92 ± 0.03 ^ab^	0.91 ± 0.02 ^ab^	0.93 ± 0.01 ^a^	0.027

Small letter “a, b, c, d” indicate a significant statistical difference between the treatments.

**Table 3 animals-13-00044-t003:** Growth performance of olive flounder *Paralichthys olivaceus* cultured at four different initial stocking densities for 8 weeks.

Parameter	Treatment 1	Treatment 2	Treatment 3	Treatment 4	*p*
Initial biomass (kg)	2.97 ± 0.11 ^d^	4.37 ± 0.11 ^c^	6.44 ± 0.06 ^b^	7.73 ± 0.17 ^a^	0.000
Initial stocking density (kg/m^2^)	3.29 ± 0.13 ^d^	4.84 ± 0.12 ^c^	7.14 ± 0.06 ^b^	8.56 ± 0.19 ^a^	0.000
Final biomass (kg)	6.93 ± 0.55 ^d^	11.98 ± 0.40 ^c^	18.20 ± 1.11 ^b^	20.88 ± 0.61 ^a^	0.000
Final stocking density (kg/m^2^)	7.68 ± 0.61 ^d^	13.28 ± 0.44 ^c^	20.16 ± 1.23 ^b^	23.14 ± 0.68 ^a^	0.000
Feed conversion	0.98 ± 0.16	0.92 ± 0.09	0.99 ± 0.03	0.98 ± 0.02	0.787
Specific growth rate (%/day)	1.46 ± 0.11 ^b^	1.72 ± 0.07 ^ab^	1.75 ± 0.03 ^a^	1.67 ± 0.03 ^b^	0.020
Daily feed intake rate (%/day)	1.41 ± 0.15	1.58 ± 0.20	1.73 ± 0.05	1.63 ± 0.02	0.071
Survival rate (%)	96.0 ± 0.0	96.0 ± 3.5	97.3 ± 3.8	97.3 ± 1.2	0.858
Condition factor	0.97 ± 0.02	0.98 ± 0.06	1.02 ± 0.05	1.02 ± 0.01	0.331

Small letter “a, b, c, d” indicate a significant statistical difference between the treatments.

**Table 4 animals-13-00044-t004:** Results of whole-body proximate composition assessment of olive flounder *Paralichthys olivaceus* cultured at four different initial stocking densities for 8 weeks.

Parameter	Treatment 1	Treatment 2	Treatment 3	Treatment 4	*p*
Moisture (%)	72.7 ± 1.2	73.6 ± 0.8	72.8 ± 0.3	72.8 ± 1.8	0.723
Crude ash (%)	3.43 ± 0.09	3.71 ± 0.18	3.61 ± 0.23	3.67 ± 0.19	0.287
Crude fat (%)	3.84 ± 1.05	4.24 ± 0.60	4.56 ± 1.07	4.14 ± 0.63	0.786
Crude protein (%)	19.9 ± 0.6 ^b^	19.0 ± 0.5 ^b^	19.7 ± 0.6 ^b^	16.9 ± 1.2 ^a^	0.005

Small letter “a, b” indicate a significant statistical difference between the treatments.

## Data Availability

The data that support the findings of this study are available from the corresponding author upon reasonable request.

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
