# Peer review of "Does Stocking Density Affect Growth Performance and Hematological Parameters of Juvenile Olive Flounder Paralichthys olivaceus in a Recirculating Aquaculture System?"

_animals, 2022, doi:10.3390/ani13010044_

Round 1

Reviewer 1 Report

The information on IGF and other blood parameters is interesting and the culture system, although not widely used for the culture of flounder is one that will become more common in the future. As such, the information reported in this study in important. 

Author Response

Thank you for your generous comments and we changed some contents and English of the manuscript.

Once again, thank you for your interest in our manuscript.

Author Response

Thank you for your interest in our manuscript.

The answers to your comments are attached as a file. 

Reviewer 3 Report

This is a nice manuscript, however I have some comments and questions below.

L90 All 12 tanks were in one RAS. How would this have interfered with water quality? In the higher stocking density, we would have at least more nitrate? It has been proven that cortisol can accumulate in the water, thus interfering in stress responses for different stocking densities, how did you deal with this problem? Or you have not considered it in your analysis/discussion?

L140 Were the fish fasted for 6 days around 4 weeks measurements, 3+3? How did the fasting time influence growth and stress parameters? Why were fish fasted for this period?

Were fish fasted for 3 days at the end of the experiment as well?

L161 if blood was centrifuged, perhaps it should be plasma and not serum

L176 – 180 Shouldn’t homoscedasticity be checked before ANOVA?

L192 – daily what?

L192-194 I didn’t understand, was there a difference or not? 

I cannot find all information in Table 3, where length is informed? Not in the text as well.

Why was feed efficiency better?

How do you explain IGF 1 was decreased, and cortisol increased, if G is similar for the high stocking density?

Figure 2 You mention changes in GH in the caption, but there is no difference informed in the graph. What is correct, the graph or the caption?

Figure 4 You also mention changes in the caption, but there is no difference informed in the graph. What is correct, the graph or the caption?

L291-294 In this case (ref 35), which stocking density had better performance?

L364 If I understood it well, growth performance and cortisol were impaired at the highest stocking density. So, If I’m correct, here you should say 20kg/m2, or not?

Final remarks

The authors mention in the introduction, that high stocking density can improve profitability due to larger production. The final biomass is increased at the higher biomass, but this is not discussed. At what point the higher biomass could be good for the fish farmer, even at the cost of higher cortisol level for example?

A deeper discussion regarding welfare and fish production should be included and included in the conclusion.

Author Response

(The authors gave the same response as above.)

Round 2

Reviewer 3 Report

The manuscript can be published as it is presented.